# Modeling and Position Control of Fiber Braided Bending Actuator Using Embedded System

Mohd Nizar Muhammad Nasir [1], Ili Najaa Aimi Mohd Nordin [1,*], Ahmad Athif Mohd Faudzi [2,3], Mohamed Naji Muftah [2], Mohd Akmal Mhd Yusoff [4] and Shahrol Mohamaddan [5]

1 Department of Electrical Engineering Technology, Faculty of Engineering Technology, Universiti Tun Hussein Onn Malaysia, Pagoh 84600, Malaysia

2 Faculty of Electrical Engineering, Universiti Teknologi Malaysia, Skudai 81310, Malaysia

3 Centre for Artificial Intelligence and Robotics, Universiti Teknologi Malaysia, Kuala Lumpur 51400, Malaysia

4 Engineering Research Centre, MARDI Headquarters, Persiaran MARDI-UPM, Serdang 43400, Malaysia

5 Department of Bioscience and Engineering, College of Systems Engineering and Science, Shibaura Institute of Technology (SIT) Fukasaku 307, Saitama 337-8570, Japan

* Correspondence: ilinajaa@uthm.edu.my

**Abstract:** The System identification (SI) black box method is used in this study to obtain the mathematical model of a fiber braided bending actuator (FBBA) using MATLAB Simulink. Data from the system input and output are used by the black box method. Thus, the voltage supplied to the electro-pneumatic regulators and the position (angle) of the FBBA system are used to collect input–output data in this study. In the system, PRBS generators are used to generate an input signal for the electro-pneumatic valve. The auto-regressive with exogenous input (ARX) model is chosen. As the controller for the FBBA position system, PID with the Genetic Algorithm (GA) tuning method and auto-tuned tuning method is proposed. The reference angle, simulation, and actual test are compared. The mathematical model gained from the SI method is verified through the simulation and test result of the position control. It was found that the model obtained through SI able represent the actual plant.

**Keywords:** system identification; soft actuator; ARX; fiber braided bending actuator; Arduino

## 1. Introduction

Soft actuators have grown in popularity among researchers and even in the industry over the years [1]. Manufacturing, manipulation, gripping, human–machine interaction, locomotion, and many other applications are covered by soft actuators [2]. This is due to the benefits they have over-rigid actuators. Soft actuators are more compliant, less expensive to produce, can be used in structured and unstructured situations [3], are lightweight, feature an increased power-to-weight ratio, and are easier to access than their counterparts [4]. McKibben muscle, balloon type [5,6], elastomeric-based [7,8], and many other approaches have been developed for the construction of soft actuators. McKibben muscle is a standard Pneumatic Muscle (PM) with an elastic cylinder tube and a double helical braid draped on the outside that is commonly used in current soft rehabilitation robots [9]. McKibben muscle has gained popularity due to its practical properties, including large contraction strains, high blocked forces, and short response times [10].

Soft actuators can generate a variety of movements, including bending [11], rotary [12], contraction, expansion [13], and have been studied and fabricated. Soft actuators have demonstrated execution tasks such as grasping and gripping objects [14], and generating various crawling and gait movements [15]. V. Oguntosin's article focuses on a soft actuator that generates a rotary motion, where the soft actuator's expansion and contraction produce clockwise and anti-clockwise rotary motions that are linked to a joint with an antagonist and agonist pair. B. K. Johnson et al. used hydraulically amplified self-healing electrostatic

(HASEL) actuators in a system, with each foldable HASEL actuator producing an extraction and contraction motion [16]. Despite the benefits, the nonlinearity of soft actuators poses challenges in modelling, controlling, and achieving fast response times [17].

Because of the system's nonlinearity, open-loop systems are not suitable for soft actuator applications. According to S. Zhong et al. pneumatic actuator muscles with open-loop control frequently struggle with high-precision control [18]. To overcome the disadvantages, a closed-loop control system is implemented. A closed-loop control strategy developed by T. G. Thuruthel is able to demonstrate a dynamic task while improving accuracy, robustness, and the actuator's conformance to the environment [19]. The system identification technique is one of the ways to realize the idea of implementing closed-loop control through a transfer function that mimics the actual actuator. The consistency of a response system is calculated straight from its transfer feature [20]. Soft robots can collect data throughout their operational range, making them ideal for data-driven system identification techniques [21].

In this study, the system identification was performed using a black box method. The black box method for system identification can be described as a blind approach [22]. This method is gaining popularity because it does not necessitate in-depth knowledge to develop the mathematical model that replicates the actual plant [23]. The system is defined based on observed behaviour in order to build data-driven models that use input and output data from an open-loop experimental setup. Controlling a fiber-reinforced soft bending actuator (FRSBA) employs the black box method [24]. Soft actuators are popular for their nonlinearity properties [25,26]. To overcome this downside, the implementation of controllers can overcome the nonlinearity of a soft actuator system [27–29]. A wide selection of controllers have been introduced over the years that implicate nonlinear control, linear control, and artificial intelligence methods, for instance, PID [30,31], Model Predictive Control (MCP) [32,33], Sliding Mode Control (SMC) [34,35], and adaptive control [36,37].

This study aims to develop a real-time position model for a fiber braided bending actuator (FBBA) developed by I.N.A.M. Nordin et al. [38–42] and implement a control algorithm to improve its performance. The SI black box method is proposed to obtain the position dynamic behaviour of the FBBA and then a PID controller is used to achieve the control objective. The PID controller is chosen due to its ease, dependency, strength, robustness, and high percentage of implementation in the industry [43–45]. However, one of the main challenges of implementing a PID controller in any system is finding the optimal values for the controller parameters ($Kp$, $Ki$, and $Kd$) [46].

To address this challenge, the study proposes using Genetic Algorithm (GA) to determine the best PID parameter values. GA is an optimization technique inspired by the process of natural selection and genetic inheritance. GA is a computational technique that iteratively searches for the optimal parameter values by evaluating a measure of quality. In this case, the quality measure is the position control performance of the soft actuator. S. A. Deraz proposed a new tuning technique that finds the best PID factors through the Genetic Algorithm (GA) method [47]. GA also gives the lowest settling time and lowest value of error when compared to Particle Swarm Optimization (PSO) and the Ziegler–Nichols Method (ZN) [30]. In this study, the GA technique is used to tune the PID controller guidelines.

Overall, the study proposes using the SI method to obtain a position model for FBBA, a PID controller to control the position, and the GA to find the optimal PID parameters. This approach can lead to improved performance of the soft actuator, which can have practical applications in various fields, including robotics, medical devices, and human–machine interfaces. The structure of this paper is as follows: The FBBA system that includes the structure of the FBBA with the experimental setup is shown in Section 2. The system identification process for obtaining the FBBA position model is discussed in Section 3. Meanwhile, Section 4 discusses the control strategy that will elaborate more on PID and GA tuning. In Section 5, the results of the simulation and real experiment used to evaluate the performance of the FBBA are presented, followed by a discussion on the FBBA control system using tuned PID controllers. Finally, Section 6 presents the conclusions.

## 2. FBBA System

### 2.1. Structure of FBBA

The actuator is made up of two different angles of fiber braiding, each covering half of the cylindrical-shaped FBBA. The soft actuator is made of silicone rubber KE1603 A-B, which is weighted in a 1:1 ratio. Figure 1a shows that the fiber braided angle for extension was 90 degrees and 35 degrees for contraction. The chamber diameter of the actuator was 10 mm. The total diameter of the FBBA used was 14 mm because this configuration provides the best displacement. Figure 1b depicts the definitions of the chamber diameter and total diameter.

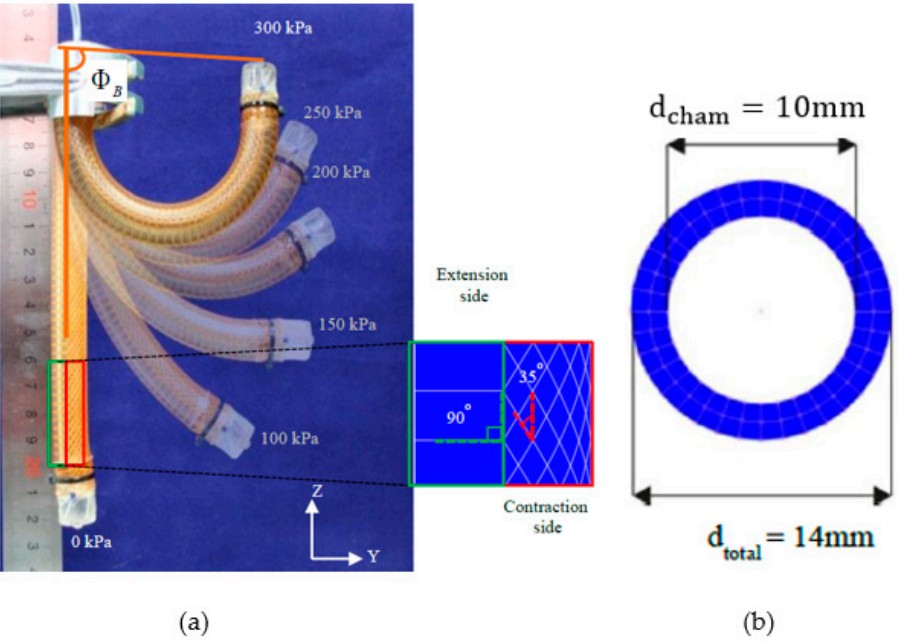

**Figure 1.** (**a**) FBBA with fiber braided angle for 90-degree extension and 35-degree contraction. (**b**) The FBBA's most optimised air chamber diameter.

### 2.2. Experimental Setup

Figure 2 depicts the experimental setup for this study. A holder stand secures the FBBA system in place, preventing it from shifting and interfering with data collection. Because the FBBA system is a single-chamber soft actuator, it has only one inlet/outlet. The FBBA's end is connected to the electro-pneumatic regulator (SMC ITV003). The MCP4725 digital-to-analogue converter (DAC) is used to provide an input signal from the Arduino to the electro-pneumatic regulator. It can accurately convert digital signals to analogue voltage signals.

The MCP4725 is a non-volatile EEPROM DAC capable of storing 12-bit data [48]. As shown in Figure 3, a flex sensor (spectra symbol FS-L-0095-103-ST) is attached to the inner bending part of the FBBA. When the actuator is actuated, it produces a bending motion on the contraction side of the actuator. The flex sensor detects the actuator's bending motion and generates a voltage value that can be translated into an angle value indicating position.

Using the experimental setup depicted in Figure 2, the black box system identification (SI) method is used to generate the mathematical model. The flex sensor is linked to the Arduino Uno, which is represented as a data acquisition (DAQ) device. The electro-pneumatic valve's supply voltage ranges from 0 V to 3 V to control the input pressure from 0 to 300 kPa.

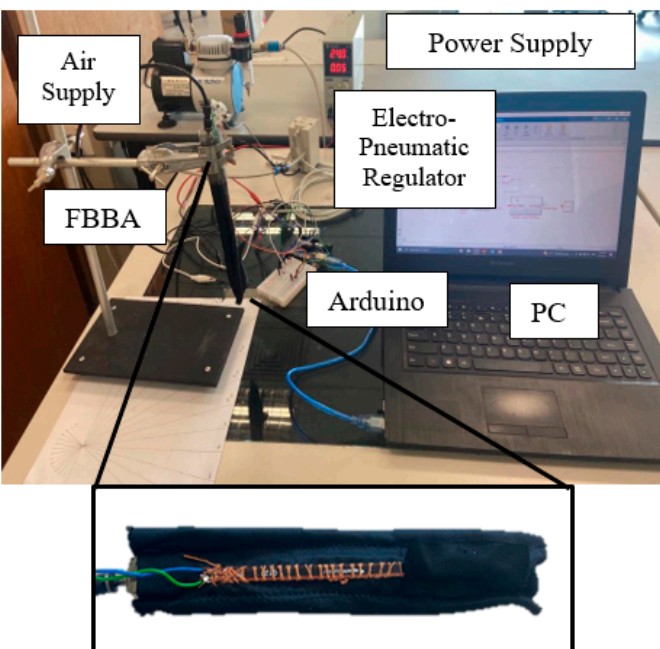

**Figure 2.** The experimental setup for the FBBA system.

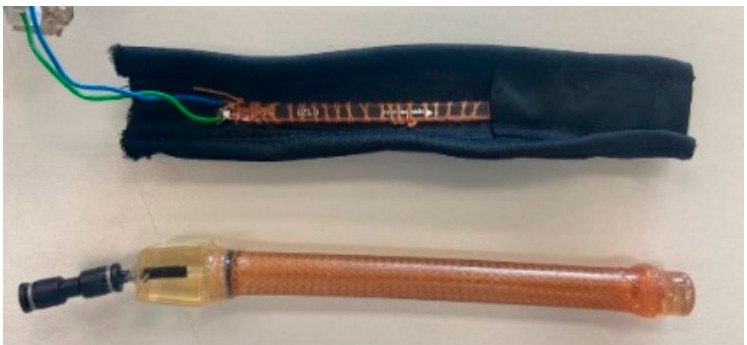

**Figure 3.** Cover to attach flex sensor (spectra symbol FS-L-0095-103-ST) to FBBA.

## 3. System Identification for Position Model

Using input–output data from an open-loop configuration, the MATLAB system identification toolbox was employed to generate the mathematical formula of the FBBA system. The experimental setup established communication between MATLAB and the actual plant by utilising an Arduino Uno as a data acquisition device. Setting up an experiment, collecting data, selecting a model set, constructing model estimation, validating the model, and selecting the mathematical model with the highest best-fit percentage are all steps in the SI procedure. The models were evaluated in a simulated plant and then in a real plant to demonstrate the validity of the mathematical formula generated through the SI approach. The input data were the voltage supplied to the electro-pneumatic valve, and the resulting data were the bending angle of the FBBA system that indicates position.

### 3.1. Model Structure Selection

When choosing a parametric model based on the applicability of the desired system, the auto-regressive with exogenous input (ARX), auto-regressive moving average with exogenous input (ARMAX), output-error (OE), and Box–Jenkins (BJ) models are just a few of the many options. The parametric ARX model is one of the popular selections for obtaining a linear mathematical model of soft actuators [49–51]. A High degree of precision has been shown in these studies. Therefore, the ARX model was selected for this study. The ARX model is represented in Equations (1) and (2), where Equation (1) is derived in the

time domain and Equation (2) is derived in the z domain. All these derivations assume that the noise is zero.

$$y(k) + a_1 y(k-1) + \cdots + a_{na} y(k-na) = b_1 u(k-d) + b_2 u(k-d-1) + \cdots + b_{nb} u(k-d-nb+1) \tag{1}$$

$$\frac{Y(z^{-1})}{U(z^{-1})} = z^{-d} \frac{B(z^{-1})}{A(z^{-1})} \tag{2}$$

where, k : discrete time step; d : time delay; na : number of poles; nb : number of zeros; u(k) : input; y(k) : output; and $z^{-1}$ : delay operator.

The ARX's structure can be determined by zeros, poles, and delay values. Different structural configurations produce different results and have a significant impact on the mathematical model's performance. ARX is represented as na − nb − nk, where na represents polynomial A order, nb represents polynomial B + 1 order, and nk represents the delay of input–output. In each system, the order can be different. Lower-order structures are commonly used because higher-order structures produce more unstable outputs and add complexity to the model.

### 3.2. Model Estimation

The quality of the frequencies used can have an impact on the outcome of parameter identification; richer frequencies provide better parameter identification. There are numerous methods for generating good input signals, including step, sinusoidal, multi-sine, pseudo-binary sequence (PRBS), and others. PRBS was chosen as the system's input signal because it can generate a random series of steps produced by the shift registers and clock. PRBS is a type of excitation signal that is widely used in system identification, but adjustments may be needed on a case-by-case basis [28]. PRBS can be created using linear-feedback shift registers (LSFR). These are made up of n flip-flops that perform an exclusive or (XOR) operation on their outputs [29]. The electro-pneumatic valve receives PRBS as an input signal, and the position of FBBA is recorded in terms of angle. A seven-bit PRBS generator block diagram was used to generate 127 random binary sequences that were fed into the electro-pneumatic regulator via an MCP4725 digital-to-analogue converter.

The input voltage generated by the PRBS signal drove the opening of the electro-pneumatic regulator, causing the FBBA to bend. The flex sensor readings were registered into MATLAB using the Arduino's A0 pin. The 7-bit PRBS was set to take 25 s to complete, with a minimum interval of 0.16 s between each random pulse.

The input–output data were split into two sections: data for estimation and data for validation. The final data collection found a total of 5001 input and output data points were collected (see Figure 4). The sampling time is also important in improving controller performance. As a result, a sampling time of 0.005 s was chosen for SI because more data can be recorded in a given time frame, influencing the model parameter estimation process [52,53]. Estimation data ranged from 0 to 2500 samples, while validation data ranged from 2501 to 5001.

### 3.3. Model Validation

The validation process determines whether or not the model obtained during the model estimation process is reliable. Equation (3) depicts the Final Prediction Error (FPE) formula, which can be used to obtain model acceptance results after analysing the best fit percentage and FPE data.

$$FPE = V \frac{(1 + n_a/N)}{(1 - n_a/N)} \tag{3}$$

The derivation of $V$ is stated in Equation (4),

$$V = \frac{e^2(k)}{N} = \frac{e^T(k).e(k)}{N} \tag{4}$$

The error vector is represented as $e(k) = [e_k \ e_{k-1} \ \ldots \ e_{k-N} \ ]^T$, $V$, as in Equation (4), is the loss function, the number of the approximated parameter is represented as $n_a$, and the sample number is represented as $N$. To obtain the most reliable transfer function, the model with the lowest FPE or Akaike's Information Criteria (AIC) is chosen from among those with different equation orders. Equation (5) depicts the AIC formula.

$$AIC = \log\left[V.\left(1 + \frac{2n_a}{N}\right)\right] \tag{5}$$

When selecting the best model generated by the system identification technique, the best fit criterion is frequently used. As shown in Equation (6), the best-fit shows the percentage of how the model was generated compared to the actual model. The greater this percentage, the greater the accuracy and precision of the simulated model.

$$fit = 100\left[1 - \frac{norm(\hat{y} - y)}{norm(y - \overline{y})}\right]\% \tag{6}$$

$$y : True\ Value$$
$$\hat{y} : Approximate\ Value$$
$$\overline{y} : Mean\ Value$$

ARX221, ARX331, and ARX441 were the three ARX models obtained with different set of orders. To obtain the best fit percentage, the ARX models were validated using validation data. According to the observations, the fourth-order model ARX441 provided the best fit with a value of 94.45%, as illustrated in Figure 5. This demonstrates the mathematical model's ability to represent the FBBA actual plant. Although ARX441 provided the best fit, ARX331 was selected with a best fit of 94.34%. This is due to the fact that higher order is more complex, and the difference in best fit between ARX331 and ARX441 was minimal. As a result, ARX331 was chosen to portray the FBBA real plant. Equation (7) depicts the transfer feature of the FBBA. Because all of the poles were within the unit circle, as shown in Figure 6, the system was considered stable.

$$G(z) = \frac{0.1135z^{-1} + 6.543z^{-2} + 15.5z^{-3}}{1 - 0.6154z^{-1} - 0.07147z^{-2} + 0.09229z^{-3}} \tag{7}$$

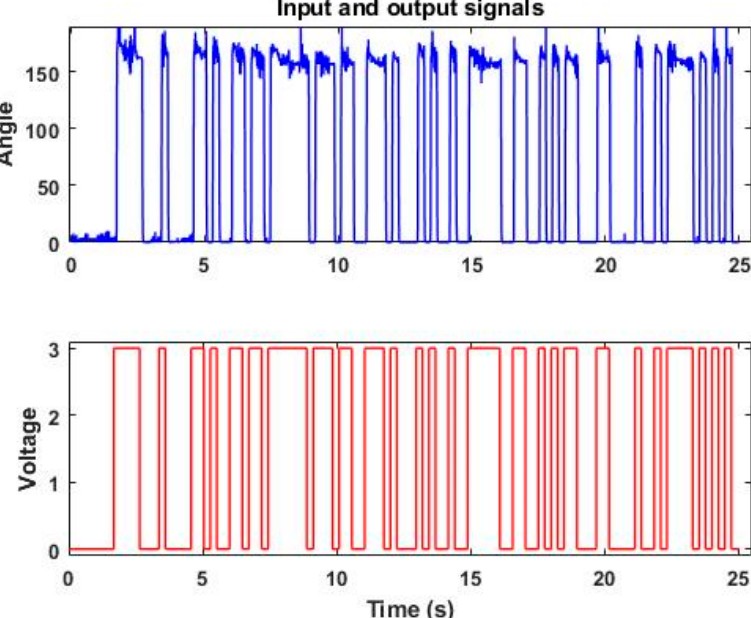

**Figure 4.** Input and output data recorded in model estimation.

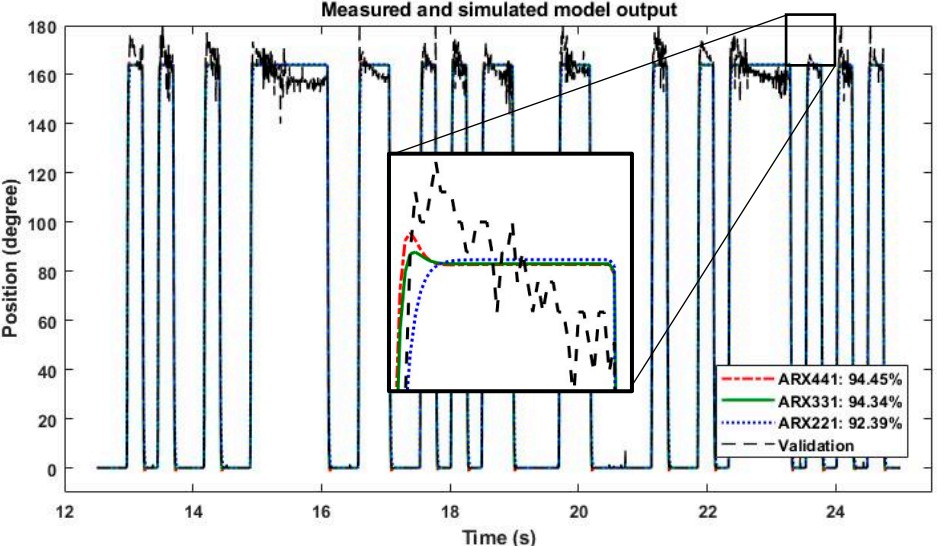

**Figure 5.** Measured and simulated model output.

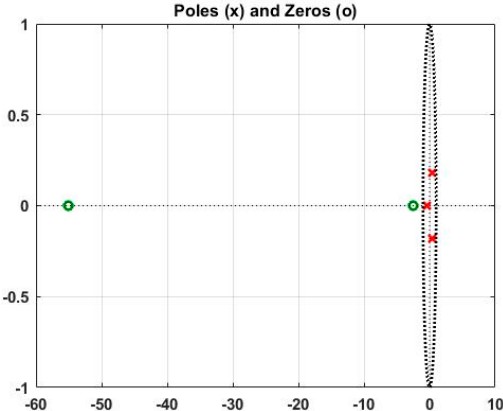

**Figure 6.** ARX331 model poles(x) and zeros (o) plot.

## 4. Control Strategy

### 4.1. PID Controller

The PID controller was used in this study to control the FBBA position and completed a closed-loop system. PID is a popular controller in the industrial sector due to its simplicity and dependability. Figure 7 depicts the discrete PID structure used in the simulation and experimentation. The study's results were determined by the optimal PID parameters $K_p$, $K_i$, and $K_d$.

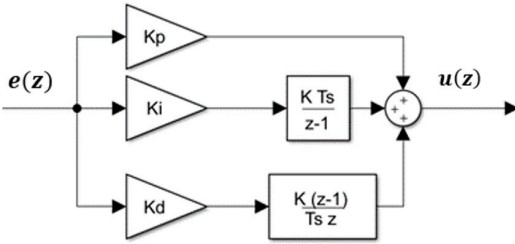

**Figure 7.** PID controller structure.

$$u(z) = \left[ K_p + K_i T_s \frac{1}{1 - z^{-1}} + \frac{K_d}{T_s} \left( 1 - z^{-1} \right) \right] e(z) \tag{8}$$

The discrete PID controller equation is shown in Equation (8). In the z domain, u(z) is the control signal and e(z) is the error signal. $K_p$ denotes the proportional increase, $K_i$ shows the integration gain, and $K_d$ symbolizes the derivative increase. Ts is the sampling time, that was set to 0.005 s.

Based on Table 1, each parameter plays an important role in providing a stable output. Therefore, the most optimal PID parameters had to be determined. Thus the Genetic Algorithm optimization technique was utilised in finding the parameters

**Table 1.** Effect of PID parameters.

| Parameter | Rise Time | Overshoot | Settling Time | Steady-State Error | Stability |
|---|---|---|---|---|---|
| $K_p$ | Decrease | Increase | Minor Changes | Decrease | Degrade |
| $K_i$ | Decrease | Increase | Increase | Eliminate | Degrade |
| $K_d$ | Minor Changes | Decrease | Decrease | No Effect | Improve |

*4.2. Genetic Algorithm (GA)*

GA is a type of evolutionarily influenced computational model. These algorithms use recombination operators to preserve critical information while encoding a possible solution to specific issues on a fundamental chromosome-like data model. GA is frequently thought of as a function optimizer. A GA implementation begins with a population of chromosomes that are typically random. The structures are then assessed, and reproductive possibilities are distributed so that chromosomes that represent an effective answer to the target issues have increased chances to develop than chromosomes that represent a weaker solution. A solution's quality is typically defined in terms of its current population.

The flow of the standard GA procedure is illustrated in Figure 8.

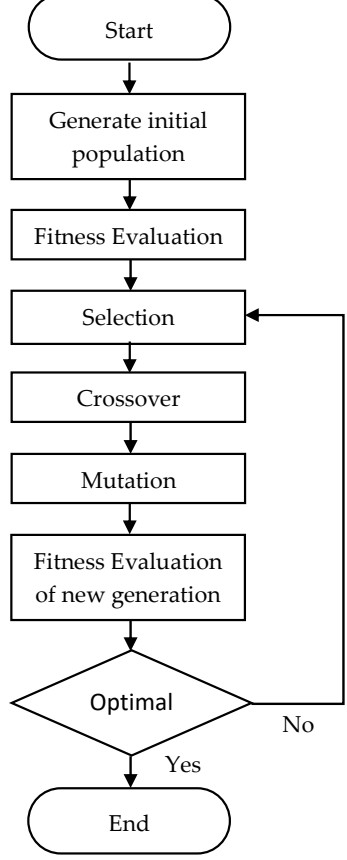

**Figure 8.** Flowchart of the standard GA process.

The most important phase in using GA is determining which objective functions will be utilized to assess the fitness of every chromosome. The objective function has various options such as the Mean of Squared Error (MSE), Integral of Time multiplied by Absolute Error (ITAE), Integral of Absolute Magnitude of Error (IAE), and Integral of Squared Error (ISE). ITAE was utilized in this study to evaluate the chromosome, as shown in Equation (9).

$$ITAE = \int_{o}^{t} t|e(t)|dt \tag{9}$$

Table 2 lists the parameters used in this study for the GA process. With a population of 30, 100 iterations were run. The optimal PID guidelines were acquired towards the conclusion of the GA process.

**Table 2.** Genetic Algorithm (GA) parameters.

| Parameter | Population Size | Iteration | Selection | Crossover | Mutation |
|---|---|---|---|---|---|
| Settings | 30 | 100 | Tournament | Arithmetic | Adaptive feasible |

*4.3. Auto-Tuning Method*

MATLAB provides several methods for tuning PID controllers, including graphical methods, automated tuning tools, and manual tuning options. M provides several graphical tools for analysing the response of a closed-loop system and determining the appropriate PID parameters. The most common graphical methods include the step response method, which involves plotting the system's step response and adjusting the PID parameters until the response meets the desired specifications, and the frequency response method, which involves analysing the system's Bode plot and adjusting the PID parameters to achieve the desired gain and phase margins. Moreover, MATLAB provides several automated tuning tools for PID controllers, which use various algorithms to determine the optimal PID parameters based on the system's response to a test input. The most common automated tuning tools in MATLAB include the PID Tuner, which uses the internal model control (IMC) method to optimize the PID parameters.

IMC is a method that can be used to optimize the parameters of a PID controller. The IMC method is a model-based control technique that involves using a mathematical model of the process being controlled to design the controller. The first step in the IMC method is to identify a mathematical model of the process being controlled. This model should accurately describe the dynamics of the process over a wide range of operating conditions. Once the process model has been identified, the next step is to design the controller using the IMC method. The controller is designed by first selecting a suitable set of performance specifications, such as the desired closed-loop response time, overshoot, and settling time. Then, the controller parameters are selected based on the model of the process and the performance specifications. The final step in the IMC method is to tune the controller to ensure that it meets the desired performance specifications. This involves adjusting the controller parameters until the desired closed-loop response is achieved.

The IMC method can be used to optimize the parameters of a PID controller by incorporating the process model into the controller design. The process model can be used to design a controller that is better suited to the dynamics of the process, resulting in better performance and stability. By using the IMC method to optimize the PID parameters, it is possible to achieve better control of the process, reduce variability, and improve product quality.

**5. Results and Discussion**

This section discusses the simulation using the ARX331 transfer function and a real-time experiment utilizing the FBBA. The PID parameters obtained from GA were tested in a simulation and a real-time experiment. The performances of the PID tuned by GA

were compared with the PID tuned by the auto-tuner toolbox and the parameters were evaluated.

### 5.1. Simulation Results

Simulink was used to test the PID-GA parameters in the simulation. The block diagram used to simulate the system is shown in Figure 9. The ARX331 transfer function obtained during the system identification process is designed to mimic the FBBA real plant. A 100-degree step source was used as the reference angle, and the controller's performance was measured in terms of overshoot (OS %), rise time (Tr), and settling time (Ts). The simulations were run with a sampling time of 0.005 s.

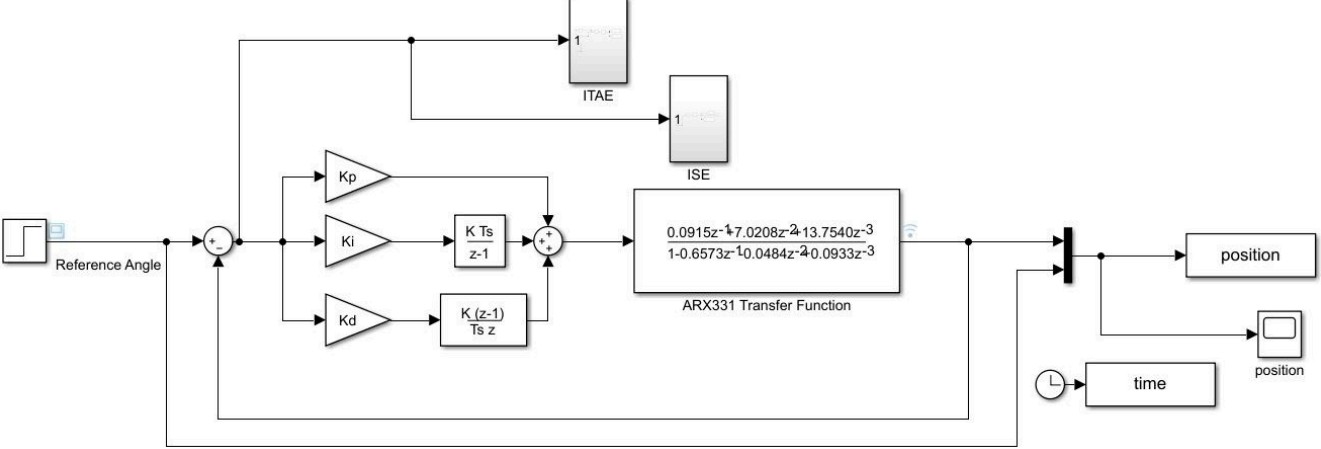

**Figure 9.** Simulation block diagram.

Table 3 summarises the PID parameters used in the simulation. The parameters were as follows: $K_p$, $K_i$, and $K_d$.

**Table 3.** Values of PID parameters.

| PID Variables | PID Auto-Tuned | PID-GA |
|:---:|:---:|:---:|
| $K_p$ | 0.0002 | $4.78 \times 10^{-9}$ |
| $K_i$ | 0.02 | 0.03 |
| $K_d$ | $5.00 \times 10^{-7}$ | $2.986 \times 10^{-7}$ |

Figure 10 compares the simulated closed-loop step response of the PID-auto-tuned and PID-GA performance. A reference angle with a constant value of 100 degrees was set as the desired position. From the result, PID-GA shows a better overall performance when compared to PID-auto-tuned.

Table 4 summarizes the controllers' performance based on the rise time (Tr), settling time (Ts), and overshoot (OS %). Both controllers produced a value of 0 for OS %, but in terms of Tr and Ts, the PID-GA controller produced a better result with 1.26 s and 2.29 s, respectively.

**Table 4.** Simulations results.

| Controller | Tr (s) | Ts (s) | OS % |
|:---:|:---:|:---:|:---:|
| PID-auto-tuned | 2.12 | 3.81 | 0 |
| PID-GA | 1.26 | 2.29 | 0 |

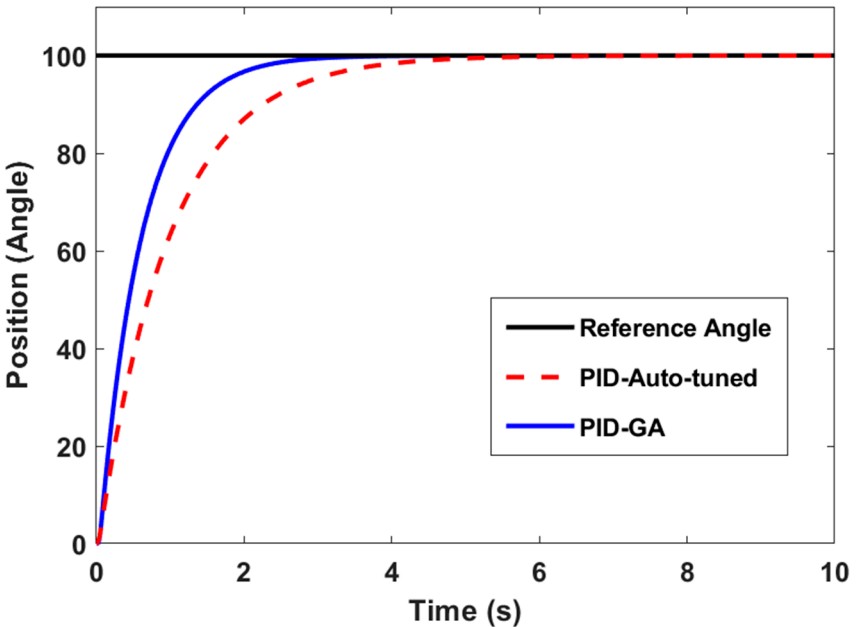

**Figure 10.** Step response of PID-auto-tuned and PID-GA.

### 5.2. Real-Time Experiment Results

The FBBA was controlled in real time by using Arduino as a communication medium between Simulink and the real plant. The program consisted of a two block diagram. Figure 11 shows the block diagram that was built into the Arduino hardware and Figure 12 shows the block diagram that was run by simulation in the Simulink software. The block diagram used in the program creates a connection between Simulink software and Arduino hardware.

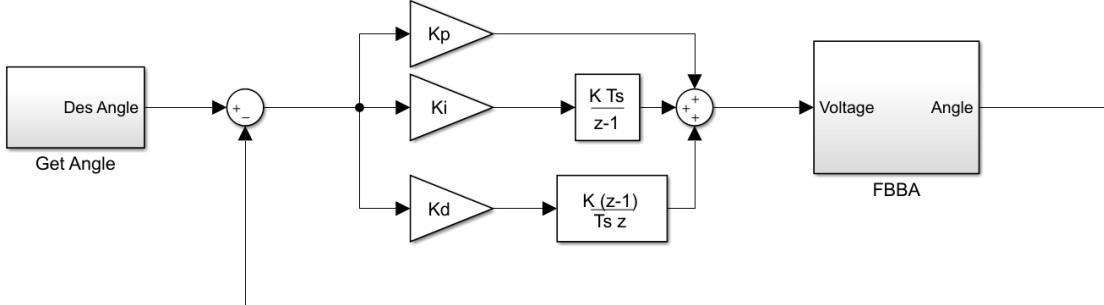

**Figure 11.** The block diagram that is embedded into Arduino.

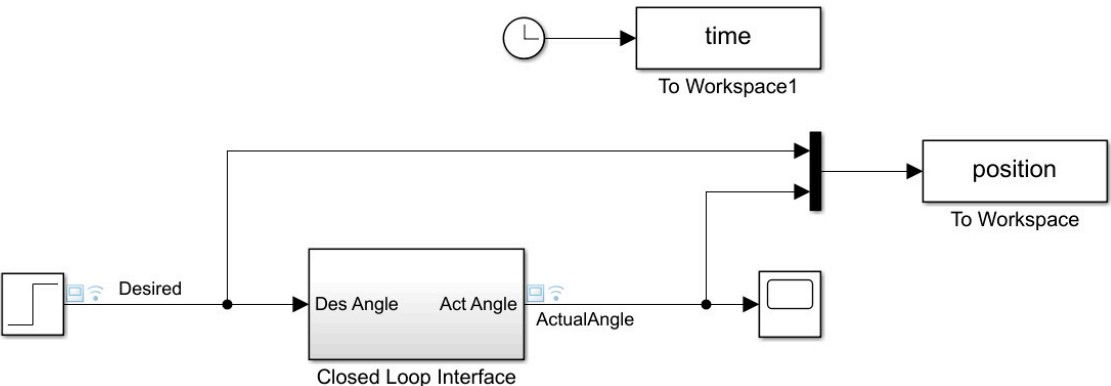

**Figure 12.** The block diagram used in simulation to manage FBBA in real time.

In the experiment of real-time control, the same PID parameters as in the simulation were used. Figure 13 depicts the step response. The experimental result follows the simulation pattern, with PID-GA outperforming PID-auto-tuned in terms of Tr and Ts. A multistep reference angle, as shown in Figure 14, was used as the input signal to evaluate the controller's effectiveness. This clearly demonstrates that PID-GA improved significantly. The study found that the system had a low to no OS %.

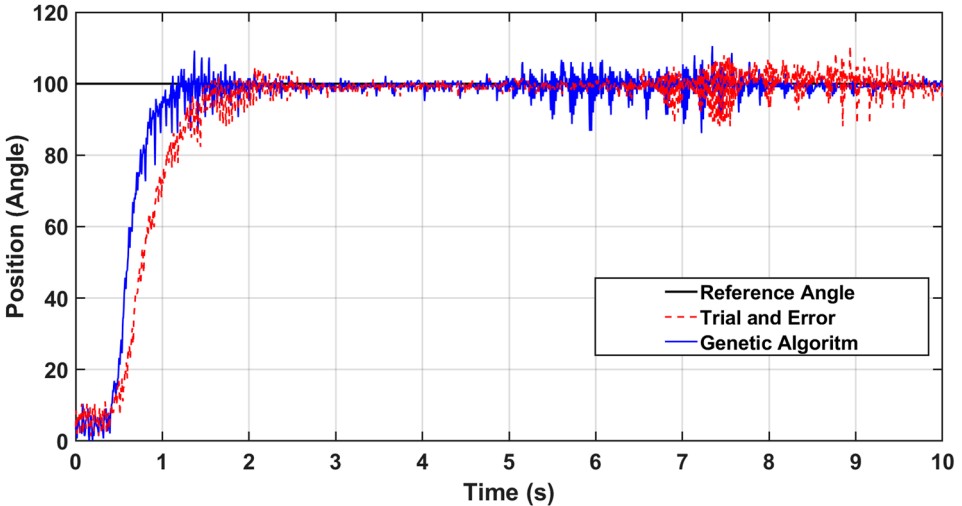

**Figure 13.** FBBA in real-time experiment step response.

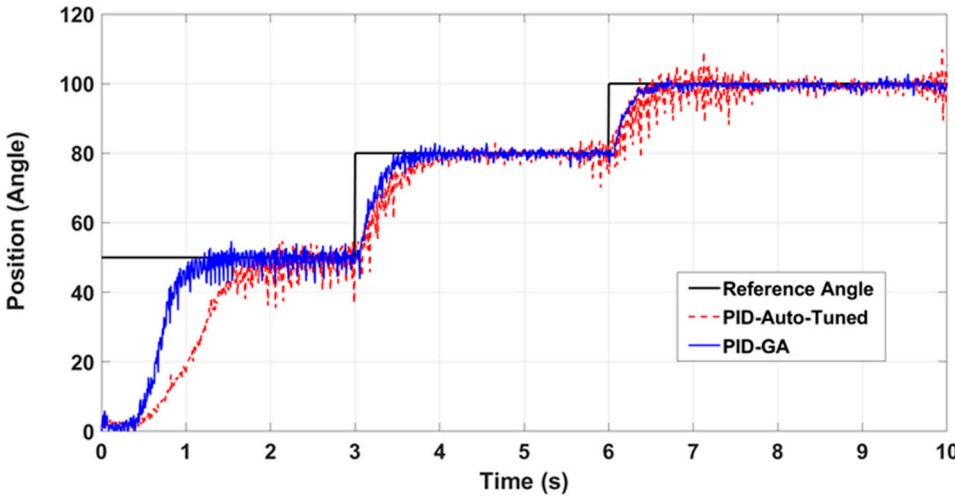

**Figure 14.** FBBA in real-time experiment multi-step response.

## 6. Conclusions

The SI black box method was used in this paper to develop the FBBA mathematical model. The SI approach was used in MATLAB software, and the model structure ARX331 was chosen as the final mathematical model for the actual plant, with a best fit value of 94.34%. To validate the model estimated from the SI black box method, the PID controller was used in both the simulation and experimental setup. The auto-tuned and GA methods were used to tune the PID parameters. The GA tuning method outperforms the other methods in simulation, with a rise time of 1.26 s, a settling time of 2.29 s, and an overshoot percentage of 0%. The actual experiment results show that PID-GA follows the simulation pattern, with better controller performance based on the rise time and settling time. In conclusion, the mathematical model representing the FBBA system was successfully developed, and the FBBA position control using PID was implemented in a real-time setup.

**Author Contributions:** Conceptualization, A.A.M.F. and M.N.M.; methodology, M.N.M.N., A.A.M.F., M.N.M., M.A.M.Y. and S.M.; software, M.N.M.N., M.N.M. and M.A.M.Y.; validation, I.N.A.M.N.; formal analysis, A.A.M.F.; investigation M.N.M.N.; resources, I.N.A.M.N.; data curation, M.N.M.N.; writing—original draft preparation, M.N.M.N.; writing—review and editing, I.N.A.M.N., A.A.M.F. and M.N.M.; visualization, A.A.M.F.; supervision, I.N.A.M.N. and A.A.M.F.; project administration, I.N.A.M.N.; funding acquisition, S.M., A.A.M.F. and I.N.A.M.N. All authors have read and agreed to the published version of the manuscript.

**Funding:** The authors would like to acknowledge the sponsor provided by Ministry of Higher Education Malaysia (MOHE) through support under Fundamental Research Grant Scheme (FRGS/1/2019/TK04/UTM/02/41) and FRGS RACER (RACER/1/2019/TK04/UTHM//8). The authors would also like to express appreciation to Universiti Teknologi Malaysia, vote no (5F137) and Universiti Tun Hussein Onn Malaysia which make this research viable and effective.

**Institutional Review Board Statement:** Not applicable.

**Informed Consent Statement:** Not applicable.

**Data Availability Statement:** Data is contained within the article.

**Conflicts of Interest:** The authors declare no conflict of interest.

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
