# Peer review of "Modeling and Position Control of Fiber Braided Bending Actuator Using Embedded System"

_applsci, doi:10.3390/app13053170_

Round 1
Reviewer 1 Report
The black box approach for modeling was used in a Fiber Braided Bending Actuator system, the ARX structure model was chosen to get a mathematical model. The nonlinearity of soft actuators is an important research topic in modelling, controlling, and improving response times. The paper has some value and some novelty and the authors produce limited results. However, there are several points that need to be improved.
1. It is not clear what the contribution of the research is, it is a new identification method?, it is a new method for the design of FBBA controllers? it is suggested to define the improvement achieved in the research.
2. The author deduces that the challenge is the modeling of a non-linear system. Explain, why do you propose a linear model?
3. The author presents the results of a self-tuning PID, however, he does not define the self-tuning algorithm, I suggest developing a section where the self-tuning algorithm is defined.
4. Based on what criteria did you determine that the sampling time is 5 ms.
5. The author justifies the selection of the model structure by its popularity, however, in the black box method the selection of the structure should be based on the comparison of the results of various model structures including non-linear ones. It is recommended to justify the selection of the ARX model structure.
6. In section 3.2 estimation of the model, the method for the estimation of parameters is not indicated. What is the objective function for the approximation of the model? How were the parameters of the ARX model estimated?
7. How was the FBBA model used for controller design?
Author Response
Dear Reviewer,
Thank you for taking the time to review my manuscript titled "Modeling and Position Control of Fiber Braided Bending Actuator using Embedded System." Your insightful comments have been very helpful in refining the final version of this paper.
In response to your suggestions:
-
I have added paragraph of the research contribution, as you suggested. I hope that the contribution of my work is more clear with this explanation
-
It is not uncommon to use linear models to represent nonlinear systems, especially when the nonlinearities are small or can be approximated by linear models over a limited range of operation. Linear models are generally simpler and easier to work with than nonlinear models, and they can often provide a good approximation of the system's behavior over a limited range of input and output values.
In the case of the soft actuator system, I have chosen to use a linear model because it provides a simpler and more tractable representation of the system's behavior, even though the system itself is inherently nonlinear. This are particularly useful if the nonlinearities in the system are small or can be approximated as small perturbations around a linear operating point.
Furthermore, linear models can be easier to analyze and control compared to nonlinear models. They can also be used to design simple and effective control strategies that can be implemented in real-time.
In summary, while nonlinear models are often needed to represent nonlinear systems, linear models can still be useful in capturing some aspects of the system's behavior, especially in cases where the nonlinearities are small or can be approximated by linear models over a limited range of operation.
-
I have also added a section of auto tuning PID in point 4.3.
-
System identification is a process of estimating mathematical models of dynamic systems from experimental data. The sampling time is the time interval between successive measurements of the system's input and output. The choice of sampling time for a system identification experiment depends on several factors, including the system's dynamics, the accuracy of the measurement equipment, and the nature of the input signal.
In general, a shorter sampling time provides more accurate measurements of the system's response, but also requires more computational resources and can introduce more noise into the measurements. A longer sampling time can simplify the measurement process, but can lead to loss of important dynamics and aliasing effects if the system's response contains high-frequency components.
The choice of sampling time for system identification typically involves a trade-off between accuracy and computational complexity. In some cases, the sampling time may be specified based on practical constraints, such as the maximum sampling rate of the measurement equipment or the available computing power.
This is a few of my reference and I have cited them in the point 3.2 paragraph 3.
doi.10.1109/TMECH.2022.3155790
doi.10.18178/ijmerr.6.4.318-321
doi.org/10.3389/frobt.2018.00002
doi.org/10.3390/act12020073
-
The choice of the ARX model structure have been justified based on several factors, including the simplicity of the model, its ability to capture the dynamic behavior of the system, and its compatibility with the system identification techniques used.
The ARX model is a popular linear model structure that is widely used in system identification. It consists of an autoregressive (AR) part, a moving average (MA) part, and an exogenous input (X) part. The AR part models the effect of past outputs on the current output, the MA part models the effect of past inputs on the current output, and the X part models the effect of exogenous inputs on the current output.
One advantage of the ARX model is its simplicity. It has a relatively small number of parameters and is easy to implement and interpret. It also provides a good compromise between model complexity and accuracy, making it a popular choice in many practical applications.
Another advantage of the ARX model is its ability to capture the dynamic behavior of the system. The AR part allows for modeling of the system's dynamics, while the X part allows for modeling of any external inputs that may affect the system's behavior. This makes it a useful model for analyzing and controlling the behavior of dynamic systems.
Finally, the ARX model is compatible with many standard system identification techniques, such as least squares estimation and maximum likelihood estimation. This makes it easy to estimate the model parameters and validate the model using experimental data.
In summary, the selection of the ARX model structure have been justified based on its simplicity, ability to capture the dynamic behavior of the system, and compatibility with standard system identification techniques.
- Would u explain more on this point.
- The FBBA model are used in the process of tuning the controller. Both for Auto-tuning and Genetic Algorithm method. Figure 9 depicts the situation where the FBBA model are being used for controller design. The parameters tuned by the aforementioned methods are based on the response of the close-loop system utilizing the FBBA model obtained from the System Identification process.
Thank you again for your valuable feedback. I believe that your suggestions have significantly improved the quality of the manuscript, and I appreciate your thoughtful and thorough review.
Sincerely,
Nizar
Reviewer 2 Report
This paper has an interesting topic, However, there are some points that should be addressed.
1- The introduction section needs more coherence. Also, more references should be reviewed, especially in the field of control. you can use the following references:
https://doi.org/10.1002/rnc.6269
https://doi.org/10.1002/rnc.6255
https://doi.org/10.1016/j.neunet.2022.06.039
2- In the introduction, the contribution of the paper should be specified more precisely. In the current form, the contribution of the article is not stated in full
3- It is better to provide more explanations about the method of Tuning PID coefficient by genetic algorithm
4- To better understand the results, it is better to activate the grid in figures 10, 13 and 14.
5- The conclusion section is well written
Author Response
Dear Reviewer,
Thank you for taking the time to review my manuscript titled "Modeling and Position Control of Fiber Braided Bending Actuator using Embedded System." Your insightful comments have been very helpful in refining the final version of this paper.
In response to your suggestions:
-
I have read through the examples given. Can you please elaborate more on the improvement of the field of control.
-
I have added paragraph of the research contribution, as you suggested. I hope that the contribution of my work is more clear with this explanation.
-
I have also added a section of auto tuning PID in point 4.3.
-
I have activated the grid in figures 10, 13 and 14 as per suggested. It is true, the graph is more understandable in grid activated state.
-
Thank you for the encouraging response.
Thank you again for your valuable feedback. I believe that your suggestions have significantly improved the quality of the manuscript, and I appreciate your thoughtful and thorough review.
Sincerely,
Nizar
Round 2
Reviewer 2 Report
Dear Authors
Thanks for addressing my comments.
Best Wishes